# Description of Calling, Courtship and Mating Behaviour of Six Species of the Genus *Zyginidia* (Auchenorrhyncha: Cicadellidae)

**DOI:** 10.3390/insects14070606

**Published:** 2023-07-04

**Authors:** Peter John Mazzoglio, Fariba Mozaffarian, Alberto Alma

**Affiliations:** 1Department of Agricultural, Forest and Food Science, University of Turin, Largo Braccini 2, Grugliasco, 10095 Turin, Italy; 2Insect Taxonomy Research Department, Iranian Research Institute of Plant Protection, Agricultural Research, Education and Extension Organization, P.O. Box 1454, Tehran 19395, Iran; faribamozaffarian@gmail.com

**Keywords:** graminicolous leafhoppers, acoustic behaviour, substrate-borne vibrations, pre-mating reproductive barriers

## Abstract

**Simple Summary:**

This paper explores the pre-mating reproductive barriers in a group of leafhopper species living vicariously in the Palaearctic region by producing hybrid zones and compares them with a closely related species of the same genus living in sympatry with some of the species of the investigated group. The results confirm the usefulness of a cheap, though neglected, methodology for recording substrate-borne vibrations and concern the basic clues that separate these species during their mating; i.e., which of the parameters of the substrate-borne vibrations they produce for calling the opposite sex and courting once the two sexes meet are necessary as pre-mating reproductive barriers. The basic behavioural isolation mechanisms are the pulse period of the female call and the main part of the courtship ritual performed by the male. An excursus into the existing literature on leafhopper bioacoustics and reproductive behaviour is undertaken to point out common aspects, propose explanations of biological phenomena—including the formation of hybrid zones, which led, in the past, to the description of several fictitious new species throughout the Palaearctic—and place the grounds for further research on pre-mating and post-mating reproductive barriers.

**Abstract:**

The vibrational communication and mating behaviour of the graminicolous leafhoppers *Zyginidia pullula*, *Z. ribauti*, *Z. scutellaris*, *Z. serpentina*, *Z. sohrab*, and *Z. biroi* were investigated to explain why the first five species hybridize. *Z. biroi* was used as a control species. All species behaved in the same way and no significant statistical differences were detected with regard to male calls, while female calls and the male courtship song differed in *Z. biroi*, thus showing that a specific pre-mating isolation mechanism was used by the latter species and the first five ones lacked such a mechanism. In addition, *Z. sohrab* is missing in Italy, while the other species live allopatrically in Italy, with the only exceptions being *Z. serpentina* and *Z. biroi*, which live in Sicily and are often found in sympatry, and *Z. scutellaris* and *Z. biroi*, which live in Sardinia. All these species can be distinguished by means of male genital appendages; however, *Z. biroi* is longer and has a different body colour. The existence of natural hybrids of *Z. pullula*, *Z. ribauti*, and *Z. scutellaris* in the Italian peninsula and their hybridization in the laboratory with *Z. serpentina* and *Z. sohrab* require the investigation of possible post-mating reproductive barriers before re-considering their systematic validity.

## 1. Introduction

Acoustic behaviour is considered the main pre-mating isolation mechanism among sympatric Auchenorrhyncha species, and substrate-borne vibrations are known to be the only means of intraspecific communication for Auchenorrhyncha, excluding Cicadidae [1,2,3,4,5]. Since the first studies [6], different species of Auchenorrhyncha have never been observed to produce the same acoustic repertoire [7]. Only in some cases of closely related species are the calls of one of the sexes and some vibrations not linked to mating purposes almost identical [8,9,10,11,12,13,14,15]. On the other hand, sibling species, which are indistinguishable on morphological grounds, have been identified according to their different acoustic production [10,16,17,18,19,20,21,22,23,24,25,26,27,28,29,30,31].

Among Auchenorrhyncha, calling signals (or common calls sensu [32]) are long-range signals associated with the initial stage of mating behaviour when the male or the female signals their presence and readiness to mate. These calls also provide information on the species and gender and further information necessary to determine the source’s location [33,34]. Males usually start calling and emit a few successive calls; then, they stop calling for a variable amount of time, walk away or jump off the plant if there is no response from a female, and, reaching a different position, produce a new series of calling signals [14,35]. These calls may act as isolating mechanisms [32] so that a species recognition mechanism is established, which will be strengthened by the courtship song.

The calling activity is an indicator of reproductive maturity and receptiveness to mating [36,37]. Pair formation starts when the female replies to a male call and a duet is established, which continues through two different behavioural stages: location and courtship [38,39]. Once the calling male receives a response from a female, the male continues to duet while searching for the stationary female and the female continues to reply once the male begins its searching pattern [40]. The male alternates calling with walking until it locates the stationary female [34,39]. During the approach stage, the continuous response by the stationary female is important for successful and quick localization by the male [34,35].

Courtship begins only when the male locates the female. The courtship is meant as behaviour aimed at increasing female acceptance [39] and, during the courtship stage, most females stop calling [34]. An unreceptive female will extend her hind leg and kick the male away; alternatively, the female may walk or jump away [32]. If the female is receptive, she raises the posterior region of the abdomen, allowing the insertion of the male’s clasping structures [41], and the pair then orient end to end and the female adjusts her wings to a position dorsal to those of the male [32,41]. Copulations generally persist for a quarter of an hour to three hours [36,39,41], and while in copula, neither the male nor female emit calls [36].

The separation of the male and female after copulation usually includes a forward movement of one or both of the individuals [41] or the female starts kicking the male with her hind legs or dragging the male away, eventually forcing the male to break genitalic contact [40].

In contrast to males, mated females stop signalling. Usually, after a period of egg-laying, the females regain their sexual receptivity [34], whereas males may start calling again as soon as they have stopped copulating.

Up to now, 11 *Zyginidia* species have been reported from Italy, all living on wild and cultivated Gramineae [42,43]. Four of these species cover the whole Italian territory in an allopatric way: *Z. pullula* (Boheman) in northern regions, *Z. ribauti* Dworakowska in central and southern regions, *Z. scutellaris* (Herrich-Schäffer) in the Aosta Valley (northwest Italy) and Sardinia, and *Z. serpentina* (Matsumura) in Sicily [44]. They can be recognized through the analysis of male gonapophyses. In 1982, males with genitalia intermediate between *Z. pullula* and *Z. scutellaris* were collected in the central Aosta Valley, where both species come into contact [43,45,46]. Subsequently, laboratory hybridization tests explained the origin of the intermediate forms found in the field [47], and studies on both the reproductive behaviour and the ethological mechanisms that lead to mating among *Z. pullula* pointed out that the production of substrate-borne vibrations induced mate recognition in this leafhopper [48]. Since acoustic and vibration signals are useful in taxonomic research on Auchenorrhyncha as a species-specific feature [24,25], the acoustic production of the four mentioned *Zyginidia* species—plus *Z. sohrab* Zachvatkin, collected in Cyprus (but also spread in the Middle East and northeastern Africa), and *Z. biroi* (Dlabola), collected in Sicily (but also spread in northern Africa and southern Spain)—was analysed in order to check if behavioural barriers preventing pair formation and, thus, interbreeding could be detected.

*Z. pullula*, *Z. ribauti*, *Z. scutellaris*, *Z. serpentina*, and *Z. sohrab* are also described in this paper as cereal-dwelling species because of their preferred host plants. *Z. sohrab* was included in this study in consideration of the fact that specimens with intermediate and aberrant aedeagi have been collected in the contact zones of *Z. pullula* and *Z. sohrab* in northern Turkey (described as *Z.* (*Z.*) *artvinicus*, *Z.* (*Z.*) *bafranicus*, *Z.* (*Z.*) *emrea*, and *Z.* (*Z.*) *karadenizicus* by Kalkandelen [49]), thus suggesting possible interbreeding between these two species [50]. In fact, interbreeding leads to the formation of intermediate aedeagi and, sometimes, the appearance of aberrations (such as bifurcations of appendages), as seen in the Turkish species. Former studies have pointed out that the five cereal-dwelling species are able to interbeed and produce viable offspring [51], and a retrospective analysis of the drawings in some papers describing *Zyginidia* species revealed hybrid facies in the contact zones of some different species; e.g., *Z. pullula* and *Z. ribauti* (see *Z. italica* Ribaut, 1947 [52] and *Z. longicornis* Vidano, 1982 [42]), *Z. ribauti* and *Z. scutellaris* (see *Z. scutellaris* in [53]), and *Z. pullula* and *Z. sohrab* (see *Z. eremita* in [54]). As far as Ribaut’s and Vidano’s collection sites are concerned, it was revealed later that they were inside a hybrid zone [51], whereas Dworakowska [53] collected the specimen she illustrated on the island of St. Honorat (Lerins Islands, Cannes, France) close to the French–Italian border in the Riviera, where hybridization occurs ([51] and Mazzoglio, unpublished data); the case of *Z. eremita* may be similar because the findings were obtained in areas where both *Z. pullula* and *Z. sohrab* may be present [51].

*Z. biroi*, in contrast, is morphologically distinguishable from the other five *Zyginidia* species mentioned because of its slightly greater length (3–3.5 mm instead of 2.5–3 mm) and its body colour, being greyish green instead of greyish blue [51]. The name *Z. biroi* was kept in spite of the synonymy [53] established with *Z. lineata* (Lindberg) living in the Canary Islands because of the differences in the shape of the aedeagus and in acoustic production (Mazzoglio, unpublished data). *Z. biroi* was chosen as a control species owing to its sympatry with *Z. serpentina* in Sicily and with *Z. scutellaris* in Sardinia and the fact that no intermediate forms between both species have ever been reported.

## 2. Materials and Methods

In vibration recording experiments prior to 1988, an accelerometer (Bruel and Kjaer 4374) was used in connection with a charge pre-amplifier (B&K 2635, l000×) and a 200 Hz high-pass filter. The signal entered the line input of a tape recorder (Sony TC-520, Sony, Japan, frequency range: 20–20,000 Hz, input sensitivity: 0.06 V, output sensitivity: 0.435 V) and was recorded at a speed of 9.5 cm/s; then, it entered a high-resolution spectrum analyser (B&K 2033) provided with a graphic recorder (B&K X-Y 2308). In the present research, a different system for the analysis of vibrations produced by leafhoppers was used following instructions given for planthoppers [55]. This magneto-dynamic system, which operates on the principle of a contact-free induction converter, consists of a coil (Bobinat; 0.9 cm in diameter and 0.6 cm in length) made of a copper wire (0.05 mm cross-section) wound 3000 times around a pressed cardboard rod (3 mm in diameter) and connected by means of a wire (B&K AO 0089; about one metre long) to the entrance of the low-impedance microphones of a Sony tape recorder (input sensitivity: 0.2 mV, output sensitivity: 0.435 V). This recording system was completed with a Samacob permanent magnet (Firma Eclipse; 4 mm in diameter, 1.5 mm in length, and 0.15 g in weight) glued onto the plant hosting the leafhoppers being tested. The coil was fixed on a rod planted in the vase so that the coil was placed near the magnet at the smallest distance without touching it.

Young and adult specimens of the six tested *Zyginidia* species were collected at the following sites: *Z. pullula*, Pessione di Chieri (province of Turin); *Z. ribauti*, Strettoia di Pietrasanta (province of Lucca); *Z. scutellaris*, Toulouse (France); *Z. serpentina* and *Z. biroi*, Dattilo di Paceco, Scopello (province of Trapani); *Z. sohrab* Larnaca (Cyprus). Rearing took place in isolation inside climatized chambers at 23 °C (+0.5) and 80% (+10) R.H. with a 16 h photoperiod, with plexiglas and nylon net cages containing the *Zea mays* L. plantlets.

For recording purposes, adults of the reared species were divided according to sex into different cages as soon as they emerged. Single individuals, couples, or groups of 3–6 specimens of the same or different sex were introduced into a plexiglas cylinder (20 cm high and 10 cm in diameter) closed on the top with a nylon net and containing a small vase with a maize plantlet at the second leaf stage and the equipment for vibration recording. The cylinder and vase were placed onto a thick layer of foam rubber and synthetic sponge in order to isolate them from floor vibrations. Males of different ages, collected directly in the field or obtained via mass rearing, were also tested. Acoustic emissions were recorded on tapes (BASF LGR 40), visualized, and successively analysed with a spectrum analyser. Later, the tape recordings were digitized and analysed statistically.

The behaviour of the tested leafhoppers was continuously noted and a detailed account accompanied each tape recording. Acoustic investigations were undertaken at room temperature (23 °C (+1)).

The length and the peak frequency of the pulses in female calls and different parts of male calls were measured by means of a high-resolution spectrum analyser. One way analyses of variance (ANOVAs) and Tukey’s tests were performed using Minitab 18.1 to compare the behavioural and acoustic parameters of the males and females of all possible paired species. *Z. biroi* was treated as a control species in the comparisons of the five cereal-dwelling species performed.

## 3. Results

*Description of the behaviour and calls of females*: As soon as females became sexually active, besides spending their time feeding and moving on the plant, they started calling spontaneously. Females always remained on the spot while calling. The pulse period (i.e., the pulse length plus the interval separating it from the following pulse) and the peak frequencies are given in Table 1 and Table 2. The call pattern is illustrated in Figure 1a for the cereal-dwelling species and Figure 1b for *Z. biroi*.

Six- to seven-day-old virgin females emitted a single call or short trains of calls followed by long periods of silence if no male answered, whereas older virgin females called uninterruptedly for days until they eventually mated. The calls were regularly spaced from each other: on average, 1.5 to 1.9 s for the cereal-dwelling species and 0.6 to 1.1 s for *Z. biroi*.

*Description of the behaviour and calls of males*: When males became sexually mature, usually a few days after the final moult, besides feeding and moving on the plant, they called erratically with a succession of pulses like very short popping sounds, especially when jumping onto a new leaf. Males were very mobile and jumped frequently off the plant. Their calls were formed of irregular sequences of pulses and intervals, as analysed in Table 3 and Table 4. The call pattern is illustrated in Figure 1c.

*Localizing the female*: Males responded to calling females by popping repetitively while tracking them; they occasionally stopped and emitted a single pop or a few intense pops. They then continued the search, repeatedly producing the lower-intensity pops until they reached the female. The low-intensity pops were formed of an irregular sequence of pulses and intervals, as shown in Table 5 and Table 6. The call pattern with the low and loud pops is illustrated in Figure 1c, and in Figure 1d, it is shown along with the female calls, forming a duet. When a male reached a female, it stopped calling, placed itself behind the female, and started courting; at this point, the female also ceased emitting calls.

*Courtship*: The first part of the courtship performed by the male was composed of about four to five irregular sequences of vibrations similar to a frog’s croaking sound. The pulse period and frequency are analysed in Table 7 and Table 8; the call pattern is illustrated in Figure 2a, and the last sequence was usually the longest one. Immediately after this prelude, the male lifted its wings straight up (Figure 3a) and started producing the second part of the courtship, which was a drumming sound for the five cereal-dwelling species (analysed in Table 9 and Table 10), the call pattern of which is illustrated in Figure 2b. The second part of the courtship for *Z. biroi* was an alternation of a pop and a croak, as shown in Figure 2c. Finally, there was a third part of the courtship, which only occurred for *Z. biroi*: a buzzing sound produced by a wing vibration by the male (Figure 2c). Courtship lasted, on average, 52.5 s for the cereal-dwelling species and 51.25 s for *Z. biroi*. After the performance of the courtship, for all species, acoustic production ceased and mating took place (Figure 3b,c). Mating durations are given in Table 11.

Two or more calling females on the same plant sometimes induced a male to track now one and then the other, eventually reaching one of them by chance. Occasionally, two or more males reached a single calling female; however, even if crowding disturbed them a little, one of the males succeeded in mating; the other(s) continued courting and sometimes attempted copulation with the couple, which usually moved away. No disturbance vibrations were produced by either sex, even in crowded conditions, but crowding caused more movements and several times the individuals jumped off the plant.

*Comparisons*: One-way analyses of variance (ANOVAs) and Tukey’s tests revealed non-significant differences in the comparisons of the female calls’ pulse periods and peak frequencies among all possible pairs of the cereal-dwelling species: *Z. pullula*, *Z. serpentina*, *Z. ribauti*, *Z. scutellaris*, and *Z. sohrab*. In contrast, they were significantly different compared to *Z. biroi* (Figure 4a–d and Table 1 and Table 2). Though the comparisons of the pulse periods of the male calls (loud popping) among all possible pairs of all tested species (including *Z. biroi*) did not show significant differences (Figure 5a,b and Table 3), the frequencies were significantly different (Figure 5c,d and Table 4), and the pulse periods and frequencies of the low popping showed significant differences (Figure 5e–h and Table 5 and Table 6). In the courtship ritual, the first part of the courtship song (croaking) did not significantly differ (Figure 6a–d and Table 7 and Table 8) among all six compared species. The second part of the courtship song (drumming)—considering the pulse period, the frequency, and the length of the performance—showed no significant differences among the five cereal-dwelling species that produced it (Figure 6e–h and Table 9 and Table 10). The mating duration showed no significant differences among all species (Figure 7a,b and Table 11).

## 4. Discussion

The technique for recording substrate-borne vibrations with planthoppers [55] is also very useful for small leafhoppers because of its very low sensitivity to airborne sound waves and the intensity of the magnetically induced impulse, which can be recorded without pre-amplification. With this recording system, low vibrations were clearly distinguishable, such as the noises made by insects while spreading brochosomes (a layer of hydrophobic proteins protecting the body [56]), running, or laying eggs, as well as the low-intensity popping emitted by males during localization of females.

Low-intensity popping was performed while the male was moving towards the female, differently from what was recorded in [34] concerning *Psammotettix alienus*, where the males signalled only when they were not walking, whereas for *Zyginidia*, it was actually the loud popping that was performed when the male stopped moving. Considering the existence of natural hybrids of cereal-dwelling species, the fact that low popping appeared to be significantly different between them suggests that it has no actual meaning for species separation. If it is perceived by the female, it could simply convey the information that a male is approaching. The differing frequencies of the loud popping among all the species could have the same meaning and may be useful only for the duet formation. The variations in the frequencies of all the substrate-borne vibrations emitted by these species, ranging from 160 to 400 Hz, could mean that they are not an important clue for pair formation, as can also be explained on a physical basis, with the result that signals produced by the same individual at different points on the same plant can vary significantly in their frequency spectra [57].

The temporal patterns and pulse periods of the calling and courtship signals are considered relevant features of the species recognition system for Auchenorrhyncha [4]. The results of the present research show that the acoustic production and mating behaviour of *Z. pullula*, *Z. ribauti*, *Z. scutellaris*, *Z. serpentina*, and *Z. sohrab* do not differ; they also confirm the results of the first investigations of *Z. pullula* [48]. Among the six tested *Zyginidia* species, the acoustic repertoire and the structure of the signals were rather simple. Only the female call consisted of regular emissions at precise intervals, whereas the male calls were always irregularly spaced. The statistical significance of the differences between these species confirms that the pulse interval and the pulse duration combine to give the overall temporal structure of the call. This signature appears to indicate species’ identity [26,27,40], and this became clear when the five cereal-dwelling species were compared with *Z. biroi*. In this comparison, the male calls and the first part of the courtship did not differ in a statistically significant manner, but the female call and the rest of the courtship song made them different, which probably also explains other similar cases reported in the literature [7].

Courtship appears to be the most complex part of the repertoire, as it is made of two (three in *Z. biroi*) acoustically and behaviourally distinct phases that are rigorously performed by the male before mating. The species-specific aspects of these signals appear to reside particularly in the second part of the courtship, which differs considerably for the five cereal-dwelling species and *Z. biroi*. The third part of the courtship may have some additional importance for *Z. biroi*; however, that could not be determined here. Buzzing produced by wing vibration during courtship has also been recorded for *Dalbulus* [32] and *Amrasca* [58]. In any case, courtship can be confirmed to be the most important species-recognition behaviour, as seen in other genera [11,32,59], and it adds to and strengthens the pre-mating reproductive barrier of the female call in the case of *Zyginidia* species when the duet is performed to enable conspecific males and females to meet [32,37,38,39,40,59,60,61].

“Satellite behaviour” was observed, as already seen for *Scaphoideus titanus* Ball [37,62,63]; i.e., a male could exploit the courtship of another male and mate with the female just prior to the attempt of the suitor. This happened once when a male performed a duet with a calling female, reached her, and performed the courtship ritual but was rejected by the female. It then waited at the side of the female, which carried on calling, and, eventually, the arrival of another male performing the courtship ritual was the occasion for the former male, waiting for the completion of the courtship ritual, to attempt copulation, this time successfully.

Considering the acoustic behaviour of leafhopper species, major affinities with *Zyginidia* are shown by *Amrasca devastans* (Distant) [58], followed by *Dalbulus* spp. [32] and *Empoasca* spp. [11]. Other genera differ much more, particularly in the greater structural complexity of the vibrations they produce [3,6,8,9,10,13,16,17,18,19,20,21,22,29,30,59,64,65,66,67,68].

Some biological remarks may help in understanding some aspects of the reproductive behaviour of the species studied. In Italy, only females of these *Zyginidia* species in the adult stage overwinter, and when they approach the winter season, they mate several times and do not lay eggs anymore. Evidence of this was given in [51] considering the fact that many of the virgin females that mated once during the acoustic recording tests were reared to obtain their offspring and the number of individuals produced was counted, giving an average number of 59 (±18). Since the highest number of offspring per overwintering female collected in the field in full winter and reared in isolation was 351, it follows that one female would need to mate up to six times in order to reach the spermatic quantity necessary for such a reproduction rate. On the other hand, males that mated once during the acoustic tests and mated a second time immediately after with another virgin female did not ever produce offspring in their second mating (nine cases tested in total), thus showing that the spermatic provision had been exhausted with the previous mating and had to be restored. The time needed for this restoration was not investigated. Therefore, a refractory period, as mentioned in [39], does not exist behaviourally in the males, but biologically it exists for sperm production.

Concerning hybrids and the studies on their acoustic behaviour in the literature, a similar but more complex case has been reported for *Dalbulus* [32]. For *Zyginidia*, the collection of hybrids in the field has been reported in the literature in the hybrid zones of the following combinations: *Z. pullula* × *Z. scutellaris* (Aosta Valley [43]); *Z. pullula* × *Z. ribauti* (Ligurian Alps and Italian northern Apennines [51,69]); *Z. scutellaris* × *Z. ribauti* (southeast France, Mazzoglio, unpublished data). Moreover, further hybrid zones are envisaged for *Z. pullula* and *Z. sohrab* in northern Turkey and probably in other contact zones for these two species in Asia. Nothing is known about northern Africa, where *Z. sohrab* is present in the east [70] and *Z. scutellaris* is present in the west [71,72]. In the laboratory, all these species have been proved to hybridize without problems and their offspring were always viable with all possible combinations and back-crossings [51]. *Z. biroi* has been used for crossings in the laboratory too but did not show any positive results with any of the cereal-dwelling species. Its sympatry with *Z. scutellaris* in Sardinia [73,74], southern Spain (Remane, pers. comm.) [53], and northern Africa [71,72], and with *Z. serpentina* in Sicily [42,73,75], can be explained by the different female calls and courtship songs, as recorded here.

On the grounds of their acoustic behaviour, the five cereal-dwelling *Zyginidia* species do not show any pre-mating isolation mechanisms, and their host plants are also the same. A more thorough behavioural analysis of the mating behaviour and choice tests are needed. The fact that *Z. pullula*, *Z. ribauti*, and *Z. scutellaris* produce hybrids in the field when they come into contact requires an investigation of the post-mating reproductive barriers among all five cereal-dwelling species before their systematic validity can be considered. We do not know how many distinct biological species of *Zyginidia* exist in Italy and, in any case, preliminary enzyme system investigations have pointed out that phosphoglucomutase (PGM) and phosphoglucose isomerase (PGI) are useful tools to discriminate between species and hybrids, yielding a finer resolution for the species complex [51]. Further studies may also include investigations of the substrate-borne vibration production and mating behaviour of *Z. lineata* from the Canary Islands in order to distinguish it from *Z. biroi* and of *Z. servadeii* Vidano, sympatric in Sicily with *Z. biroi* and *Z. serpentina*, to identify what pre-mating isolation mechanism exists for them.

## Figures and Tables

**Figure 1 insects-14-00606-f001:**
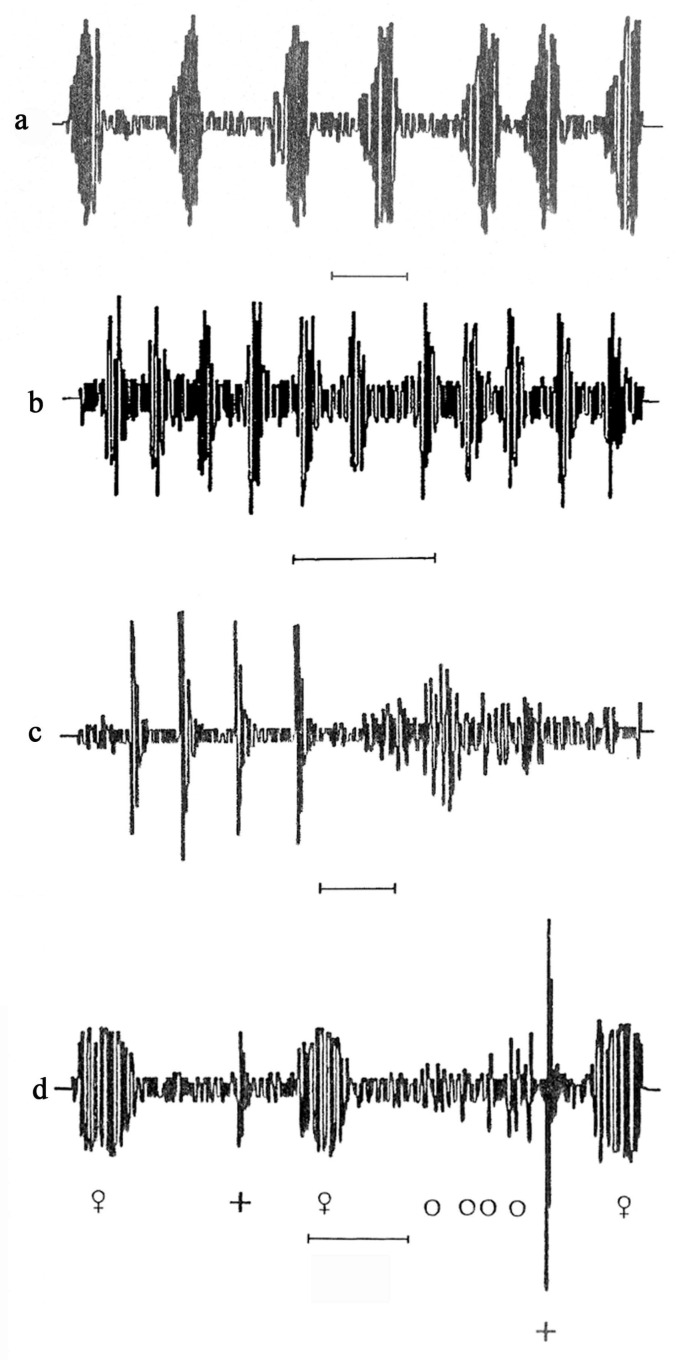
(**a**) Female call of *Z. pullula*, *Z. serpentina*, *Z. ribauti*, *Z. scutellaris*, and *Z. sohrab*; (**b**) female call of *Z. biroi*; (**c**) loud and low popping; (**d**) duet of female calls alternating with loud (+) and low (o) male calls (popping) demonstrated by cereal-dwelling species. Scale: 1 s.

**Figure 2 insects-14-00606-f002:**
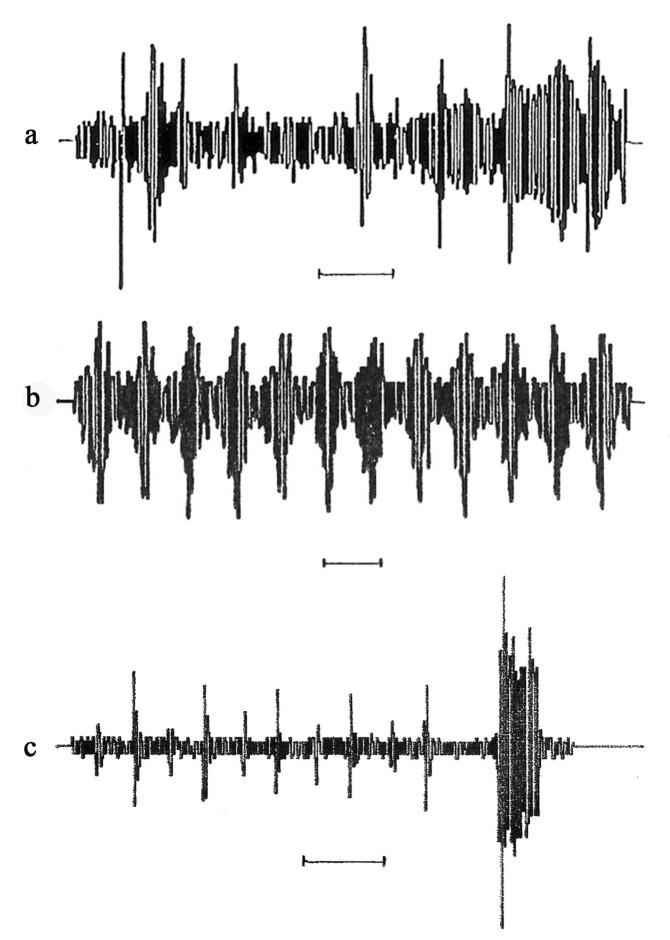
(**a**) First part of male courtship song produced by graminicolous leafhoppers: croaking for *Z. pullula*, *Z. serpentina*, *Z. ribauti*, *Z. scutellaris*, *Z. sohrab*, and *Z. biroi* (scale: 1 s); (**b**) second part of courtship song produced by cereal-dwelling species: drumming for *Z. pullula*, *Z. serpentina*, *Z. ribauti*, *Z. scutellaris*, and *Z. sohrab* (scale: 0.1 s); (**c**) second and third parts of male courtship song produced by *Z. biroi* (scale: 1 s).

**Figure 3 insects-14-00606-f003:**
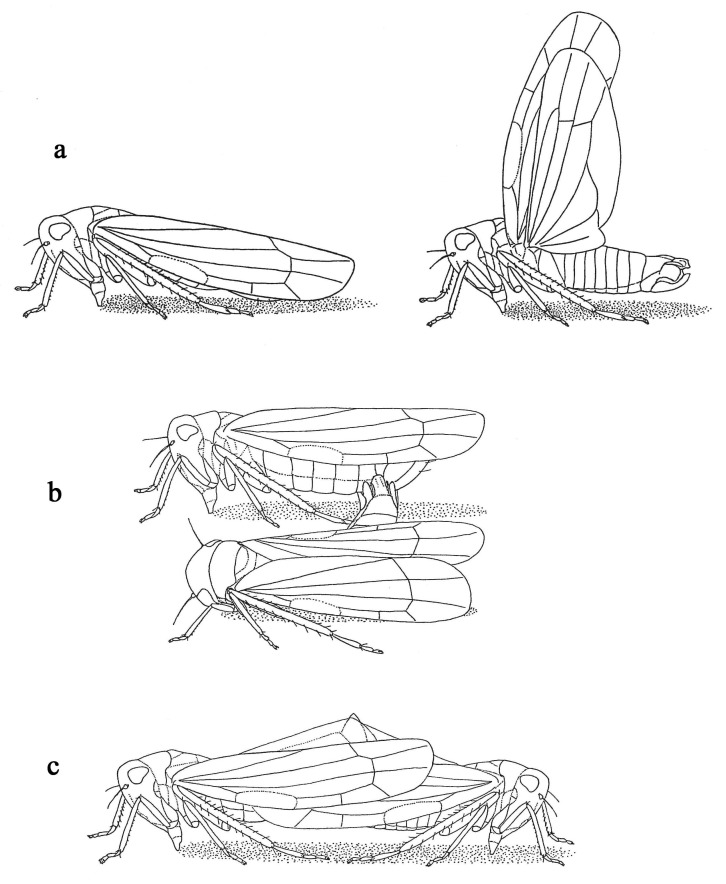
(**a**) Positions of female and male during courtship demonstrated by the six graminicolous leafhoppers studied; (**b**) mating attempt by a male; (**c**) positions of female and male while mating (ex Mazzoglio & Arzone, 1988).

**Figure 4 insects-14-00606-f004:**
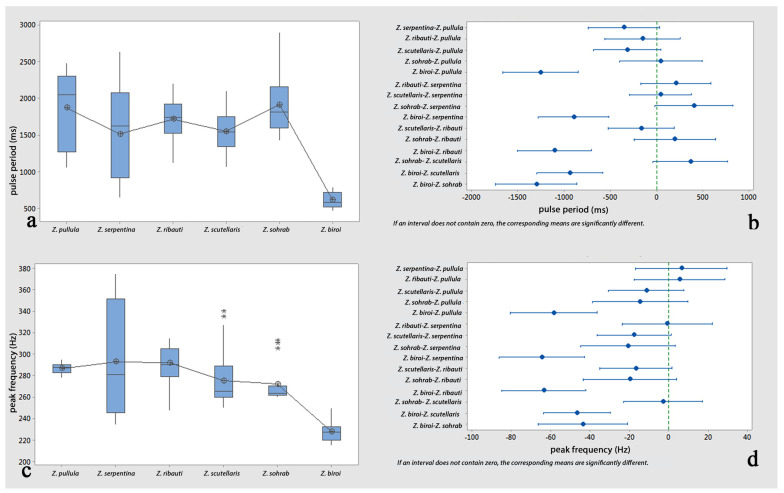
Analyses of female calls of six graminicolous leafhoppers: (**a**) comparison of pulse periods (ms); (**b**) pairwise comparisons of pulse periods (ms) using Tukey simultaneous 95% CI; (**c**) comparison of peak frequency (Hz); (**d**) pairwise comparisons of peak frequency (Hz) using Tukey simultaneous 95% CI.

**Figure 5 insects-14-00606-f005:**
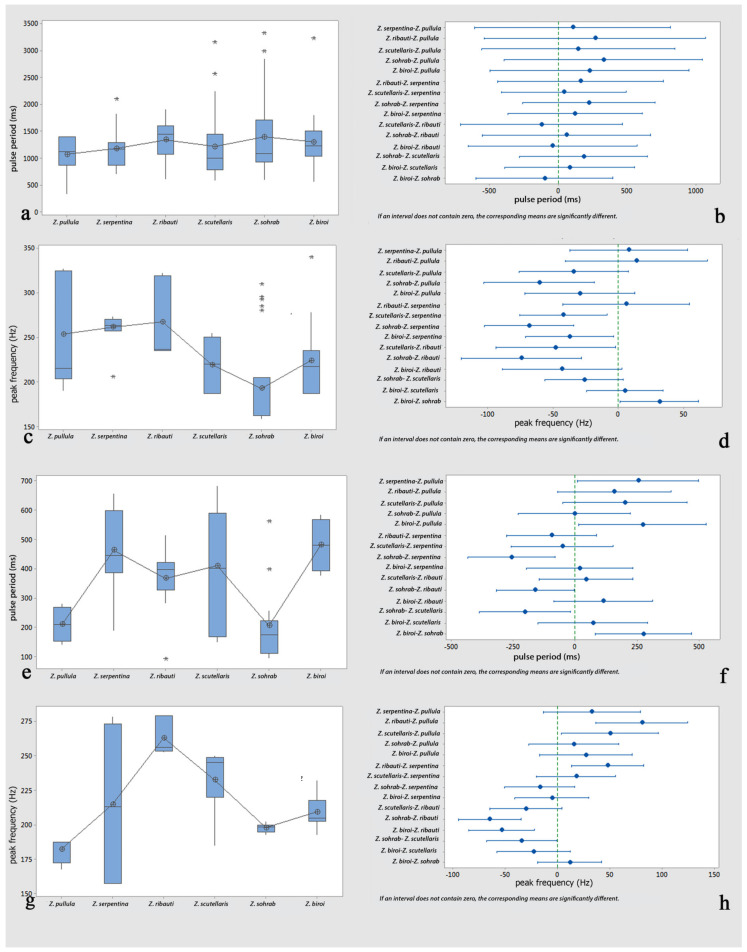
Analyses of male loud and low popping of six graminicolous leafhoppers. Loud popping: (**a**) comparison of pulse periods (ms); (**b**) pairwise comparisons of pulse periods (ms) using Tukey simultaneous 95% CI; (**c**) comparison of peak frequency (Hz); (**d**) pairwise comparisons of peak frequency (Hz) using Tukey simultaneous 95% CI. Low popping: (**e**) comparison of pulse periods (ms); (**f**) pairwise comparisons of pulse periods (ms) using Tukey simultaneous 95% CI; (**g**) comparison of peak frequency (Hz); (**h**) pairwise comparisons of peak frequency (Hz) using Tukey simultaneous 95% CI.

**Figure 6 insects-14-00606-f006:**
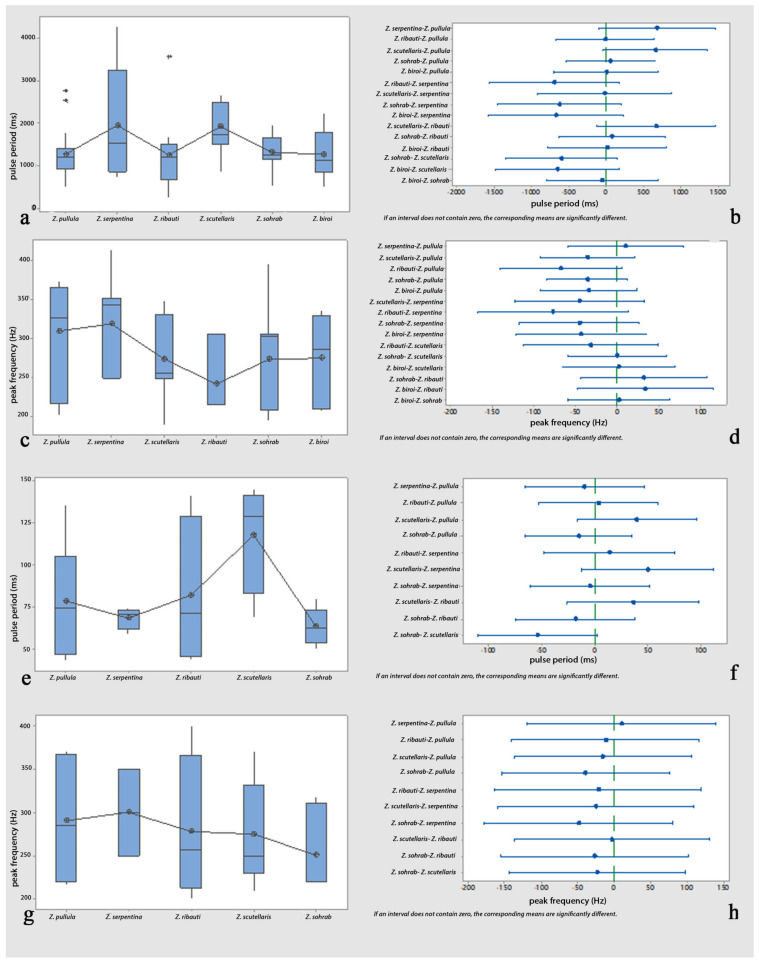
Analyses of male courtship songs of six graminicolous leafhoppers. Croaking: (**a**) comparison of pulse periods (ms); (**b**) pairwise comparisons of pulse periods (ms) using Tukey simultaneous 95% CI; (**c**) comparison of peak frequency (Hz); (**d**) pairwise comparisons of peak frequency (Hz) using Tukey simultaneous 95% CI. Drumming: (**e**) comparison of pulse periods (ms); (**f**) pairwise comparisons of pulse periods (ms) using Tukey simultaneous 95% CI; (**g**) comparison of peak frequency (Hz); (**h**) pairwise comparisons of peak frequency (Hz) using Tukey simultaneous 95% CI.

**Figure 7 insects-14-00606-f007:**
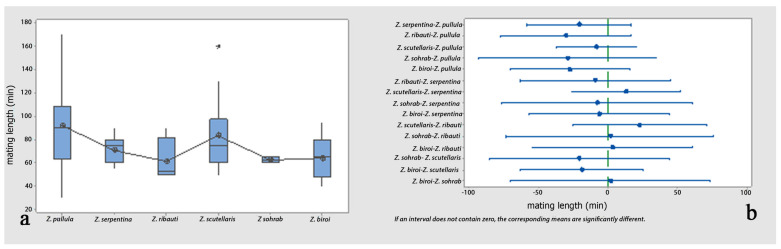
Analyses of the mating durations of six graminicolous leafhoppers: (**a**) comparison of mating durations of the six species; (**b**) pairwise comparisons of the mating durations (min) using Tukey simultaneous 95% CI.

**Table 1 insects-14-00606-t001:** Mean and Tukey pairwise comparisons of pulse periods in female calls (ms).

Species	Mean (ms)	Min (ms)	Max (ms)	StDev	95% CI	n	Grouping *
*Z. pullula*	1873	1054.7	2484.4	502	(1674, 2072)	16	A
*Z. serpentina*	1515	647	2645	625	(1341, 1688)	21	A
*Z. ribauti*	1720.1	1124.9	2203.1	281.6	(1527.0, 1913.1)	17	A
*Z. scutellaris*	1553.6	1066.4	2109.4	273.7	(1403.1, 1704.0)	28	A
*Z. sohrab*	1916	1429.7	2906.4.	408	(1686, 2146)	12	A
*Z. biroi*	614.7	465.8	797	111.3	(421.6, 807.8)	17	B

Pooled StDev = 401.503. * Grouping information using the Tukey method and 95% confidence interval. Means that do not share a letter were significantly different.

**Table 2 insects-14-00606-t002:** Mean and Tukey pairwise comparisons of female call peak frequencies (Hz).

Species	Mean (Hz)	Min (Hz)	Max (Hz)	StDev	95% CI	n	Grouping *
*Z. pullula*	286.58	277.5	295	4.50	(275.14, 298.02)	19	A
*Z. serpentina*	292.8	234	375	52.5	(281.6, 303.9)	20	A
*Z. ribauti*	291.84	247.5	315	18.38	(280.96, 302.72)	21	A
*Z. scutellaris*	275.00	250	340	23.73	(268.62, 281.38)	61	A
*Z. sohrab*	271.83	260	312.5	18.14	(259.74, 283.92)	17	A
*Z. biroi*	228.08	220	250	8.95	(218.30, 237.86)	26	B

Pooled StDev = 25.2392. * Grouping information using the Tukey method and 95% confidence interval. Means that do not share a letter were significantly different.

**Table 3 insects-14-00606-t003:** Mean and Tukey pairwise comparisons of male loud popping pulse periods (ms).

Species	Mean (ms)	Min (ms)	Max (ms)	StDev	95% CI	n	Grouping *
*Z. pullula*	1075	328.1	1406.3	383	(645, 1505)	7	A
*Z. serpentina*	1178.8	703	2109.3	369.6	(951.2, 1406.4)	25	A
*Z. ribauti*	1341	609.4	1921.9	400	(998, 1685)	11	A
*Z. scutellaris*	1218	843.7	3159.4	620	(1007, 1429)	29	A
*Z. sohrab*	1401	549.6	3336.8	778	(1164, 1638)	23	A
*Z. biroi*	1300	562.5	3234.4	566	(1057, 1542)	22	A

Pooled StDev = 574.191. * Grouping information using the Tukey method and 95% confidence intervals. Means that do not share a letter were significantly different.

**Table 4 insects-14-00606-t004:** Mean and Tukey pairwise comparisons of loud popping peak frequencies (Hz).

Species	Mean (Hz)	Min (Hz)	Max (Hz)	StDev	95% CI	n	Grouping *
*Z. pullula*	253.6	190	327.5	62.4	(228.8, 278.4)	10	A B C
*Z. serpentina*	261.48	206.2	273.7	14.33	(243.46, 279.50)	19	A
*Z. ribauti*	267.3	235	322.5	43.6	(239.6, 295.1)	8	A B
*Z. scutellaris*	226.07	172.5	287.5	36.74	(205.20, 233.41)	31	C D
*Z. sohrab*	231.61	158.7	312.5	61.39	(178.13, 207.82)	28	D
*Z. biroi*	231.17	187.5	340	34.91	(209.81. 238.49)	30	B C

Pooled StDev = 39.6711. * Grouping information using the Tukey method and 95% confidence intervals. Means that do not share a letter were significantly different.

**Table 5 insects-14-00606-t005:** Mean and Tukey pairwise comparisons of male low popping pulse periods (ms).

Species	Mean (ms)	Min (ms)	Max (ms)	StDev	95% CI	n	Grouping *
*Z. pullula*	210.1	140.6	281.1	60.1	(75.6, 344.6)	4	B C
*Z. serpentina*	462.9	187.5	375	149.2	(367.8, 558)	8	A
*Z. ribauti*	367.2	93.7	515.6	109.2	(289.5, 444.9)	12	A B
*Z. scutellaris*	409.7	149.8	683.6	199.8	(308.1, 511.4)	7	A B
*Z. sohrab*	205.9	93.7	562.5	131	(134, 277.8)	14	C
*Z. biroi*	480.5	375	585.9	92.2	(370.6. 590.3)	6	A

Pooled StDev = 39.6711. * Grouping information using the Tukey method and 95% confidence intervals. Means that do not share a letter were significantly different.

**Table 6 insects-14-00606-t006:** Mean and Tukey pairwise comparisons of low popping peak frequencies (Hz).

Species	Mean (Hz)	Min (Hz)	Max (Hz)	StDev	95% CI	n	Grouping *
*Z. pullula*	182.5	167	187.5	10.00	(156.86, 208.14)	10	C
*Z. serpentina*	215	157.5	278.5	61.2	(196.8, 233.1)	19	B C
*Z. ribauti*	262.81	252.5	278.75	11.87	(248.01, 277.61)	8	A
*Z. scutellaris*	232.5	185	250	22.83	(214.37, 250.63)	31	A B
*Z. sohrab*	197.857	192.5	202.5	3.231	(184.154, 211.560)	28	C
*Z. biroi*	209.32	202.5	210	11.84	(193.86. 224.78)	30	B C

Pooled StDev = 25.5392. * Grouping information using the Tukey method and 95% confidence intervals. Means that do not share a letter were significantly different.

**Table 7 insects-14-00606-t007:** Mean and Tukey pairwise comparisons of croaking pulse periods (ms).

Species	Mean (ms)	Min (ms)	Max (ms)	StDev	95% CI	n	Grouping *
*Z. pullula*	1271.0	515.6	2765.7	505.6	(1005.1, 1536.9)	26	A
*Z. serpentina*	1950	750	4265.6	1326	(1470, 2429)	8	A
*Z. ribauti*	1255	281.2	3562.5	828	(879, 1631)	13	A
*Z. scutellaris*	1926	867.2	2671.8	586	(1517, 2335)	11	A
*Z. sohrab*	1325.9	539	1953.2	379.8	(1006.4, 1645.5)	18	A
*Z. biroi*	1269	515.7	2248.1	656	(869, 1669)	11	A

Pooled StDev = 679.936. * Grouping information using the Tukey method and 95% confidence intervals. Means that do not share a letter were significantly different.

**Table 8 insects-14-00606-t008:** Mean and Tukey pairwise comparisons of croaking peak frequencies (Hz).

Species	Mean (Hz)	Min (Hz)	Max (Hz)	StDev	95% CI	n	Grouping *
*Z. pullula*	309.3	201.2	373.2	64.9	(287.4, 331.3)	30	A
*Z. serpentina*	318.7	248.7	413.7	62.3	(276.2, 361.2)	8	A
*Z. ribauti*	241.4	215	305	43.5	(196.0, 286.9)	7	A
*Z. scutellaris*	273.5	188.7	347.5	46.8	(241.3, 305.6)	14	A
*Z. sohrab*	273.0	195	395	64.8	(247.3, 298.6)	22	A
*Z. biroi*	274.8	207.5	336.2	56.1	(241.8, 307.8)	13	A

Pooled StDev = 60.4058. * Grouping information using the Tukey method and 95% confidence intervals. Means that do not share a letter were significantly different.

**Table 9 insects-14-00606-t009:** Mean and Tukey pairwise comparisons of drumming pulse periods (ms).

Species	Mean (ms)	Min (ms)	Max (ms)	StDev	95% CI	n	Grouping *
*Z. pullula*	78.5	43	136	34.1	(53.7, 103.3)	6	A
*Z. serpentina*	68.50	59	74	6.56	(38.16, 98.84)	4	A
*Z. ribauti*	82.0	44	141	45.0	(51.7, 112.3)	4	A
*Z. scutellaris*	117.8	69	145	33.5	(87.4, 148.1)	4	A
*Z. sohrab*	63.50	50	80	10.93	(38.73, 88.27)	6	A

Pooled StDev = 28.9871. * Grouping information using the Tukey method and 95% confidence intervals. Means that do not share a letter were significantly different.

**Table 10 insects-14-00606-t010:** Mean and Tukey pairwise comparisons of drumming peak frequencies (Hz).

Species	Mean (Hz)	Min (Hz)	Max (Hz)	StDev	95% CI	n	Grouping *
*Z. pullula*	290.4	217.5	370	78.3	(233.6, 347.2)	6	A
*Z. serpentina*	300.0	250	350	57.7	(230.4, 369.6)	4	A
*Z. ribauti*	278.1	200	400	85.6	(208.6, 347.7)	4	A
*Z. scutellaris*	274.5	210	370	60.8	(212.3, 336.7)	5	A
*Z. sohrab*	250.8	220	317.5	47.9	(194.0, 307.6)	6	A

Pooled StDev = 66.6842. * Grouping information using the Tukey method and 95% confidence intervals. Means that do not share a letter were significantly different.

**Table 11 insects-14-00606-t011:** Mean and Tukey pairwise comparisons of mating durations (min).

Species	Mean (min)	Min (min)	Max (min)	StDev	95% CI	n	Grouping *
*Z. pullula*	92.00	30	170	34.88	(78.99, 105.01)	20	A
*Z. serpentina*	70.71	55	90	12.72	(48.73, 92.70)	7	A
*Z. ribauti*	61.25	50	90	19.31	(32.16, 90.34)	4	A
*Z. scutellaris*	83.44	50	160	29.82	(68.89, 97.98)	16	A
*Z. sohrab*	62.50	60	65	3.54	(21.37, 103.63)	2	A
*Z. biroi*	64.00	40	95	20.12	(37.98, 90.02)	5	A

Pooled StDev = 28.9319. * Grouping information using the Tukey method and 95% confidence intervals. Means that do not share a letter were significantly different.

## Data Availability

The data used in this study are contained within the tables in the article.

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
