# Peer review of "Description of Calling, Courtship and Mating Behaviour of Six Species of the Genus Zyginidia (Auchenorrhyncha: Cicadellidae)"

_insects, 2023, doi:10.3390/insects14070606_

Round 1
Reviewer 1 Report
Dear authors,
Congratulations on a job well done! This is a well-conceived and meticulously researched paper, which complies with the highest scientific standard. The methods are chosen adequately to the questions, the results are presented in a concise and clear way. The present work is a valuable contribution to our knowledge on the behavioral ecology and biosystematics of leafhoppers. I have but a few minor suggestions on alternative wording which I hope will be helpful for you. Best wishes, and carry on the good work!

Author Response
Dear Reviewer,
thankyou indeed for your appreciation and help in improving our manuscript.
Your suggestions have been useful to enhance the quality of the article.
Reviewer 2 Report
You demonstrate that courtship behaviour is a major barrier for hybridisation in some Cicadellidae.
I would have some suggestions to improve your paper:
- please explain if you describe vibratory or acoustic signals
- the recordings of the sound/vibratory patterns should be presented first, before any tables are given. Ideally with an identical temporal resolution and an indication of the frequencies. The figures should be labelled to provide a better understanding of the different signal types.
The diagrams which compare data between species need a substantial revision with sub-headings and proper labelling of the axis
It is not clear if you describe pulse periods or pulse rates in the tables.
The discussion is quite lengthy and deviating from the topic of your research.
more details in the PDF

Is generally ok, but could do with some editing
Author Response
Dear Reviewer,
we thank you indeed for your valuable revision and accepted all the requests for improving the quality of our manuscript both in the technical wording and in the English language.
The manuscript has been corrected as you suggested and as can be seen in the file, thanks to the Track changes function.
Figures have been improved and placed in the order as you suggested and captions have been made more intelligible.
The discussion is slimmer according to your request and we are confident that the final result is satisfactory.
